# ViaRL: Adaptive Temporal Grounding via Visual Iterated Amplification Reinforcement Learning

## Abstract

Video understanding is inherently intention-driven—humans naturally focus on relevant frames based on their goals. Recent advancements in multimodal large language models (MLLMs) have enabled flexible query-driven reasoning; however, video-based frameworks like Video Chain-of-Thought lack direct training signals to effectively identify relevant frames. Current approaches often rely on heuristic methods or pseudo-label supervised annotations, which are both costly and limited in scalability across diverse scenarios. To overcome these challenges, we introduce ViaRL, the first framework to leverage rule-based reinforcement learning (RL) for optimizing frame selection in intention-driven video understanding. An iterated amplification strategy is adopted to perform alternating cyclic training in the video CoT system, where each component undergoes iterative cycles of refinement to improve its capabilities. ViaRL utilizes the answer accuracy of a downstream model as a reward signal to train a frame selector through trial-and-error, eliminating the need for expensive annotations while closely aligning with human-like learning processes. Comprehensive experiments across multiple benchmarks, including VideoMME, LVBench, and MLVU, demonstrate that ViaRL consistently delivers superior temporal grounding performance and robust generalization across diverse video understanding tasks, highlighting its effectiveness and scalability. Notably, ViaRL achieves a nearly 15% improvement on Needle QA, a subset of MLVU, which is required to search a specific needle within a long video and regarded as one of the most suitable benchmarks for evaluating temporal grounding.

## 1 Introduction

Recent advancements in OpenAI's o3 model (OpenAI, 2025), have demonstrated remarkable capabilities in image understanding. The model leverages multi-turn query-based grounding and powerful reasoning abilities to process visual signals alongside textual queries. Inspired by this paradigm, an intriguing question arises: *can video understanding be enhanced through a similar approach using temporal grounding?* While spatial grounding focuses on identifying key regions within an image, temporal grounding aims to pinpoint the most relevant frames in a video sequence.

It's also necessary to examine another underlying problem: *Why use the framework to identify the relevant frames rather than reasoning over complex facts?* While recent studies have attempted to enhance reasoning in video tasks, their efforts have not consistently led to improvements. Wang & Peng (2025) reports that GRPO fine-tuning results in decreased accuracy. Despite introducing a large amount of annotated reasoning data, Video-R1 (Feng et al., 2025) fails to outperform the Qwen2.5-VL-7B (without CoT/SFT/RL) on VideoMME. Similarly, the results of Team et al. (2025); Guo et al. (2025), in which compare RL and SFT models across several video benchmarks, show that RL fine-tuning does not consistently outperform SFT. These observations suggest that directly finetune a MLLM to reason about complex facts in the frames is not yet effective for general video tasks. We attribute this to two main factors: (1) most general video understanding tasks are perceptual rather than truly cognitive in nature, and (2) significant gaps remain between the physical world and MLLMs, for example, how to represent the visual information. Therefore, we adopt this framework for reasoning about relevant frames for subsequent answering, as shown in Fig 1.

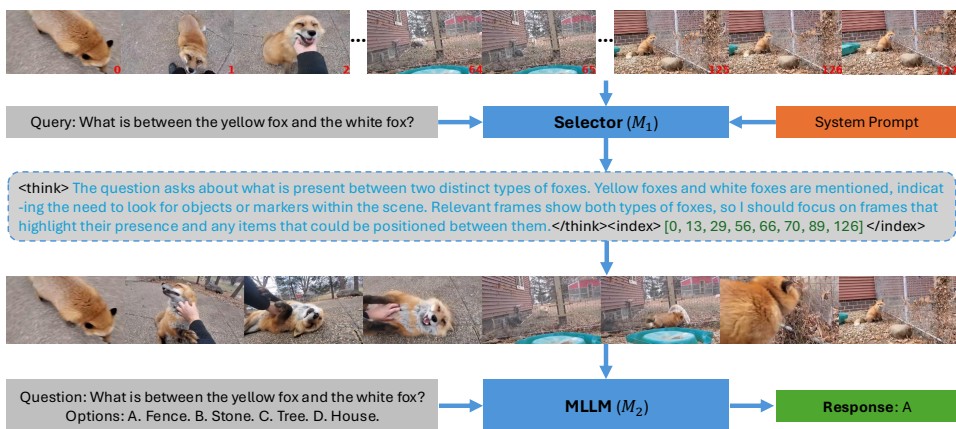

Figure 1: The overall architecture of our approach.

Several recent works have explored frame selection techniques and video Chain-of-Thought (CoT) pipelines. For instance, Hu *et al.* (Hu et al., 2025b) employs a learnable score query to predict importance scores for each frame, using pseudo-labels annotated by MLLMs during training. AKS (Tang et al., 2025) leverages the CLIP model to compute relevance scores between each frame and the query. Both of them introduce extra sophisticated strategies, such as Non-Maximum Suppression (NMS) sampling, to reduce redundancy in frame selection. CoS (Hu et al., 2025a) uses the LLaVA to evaluate whether the query elements are present in each frame to construct positive and negative shots for further co-reasoning, without temporal consideration. Additionally, Frame-Voyager (Yu et al., 2024) applies a Direct Policy Optimization (DPO) strategy to select a group from combinations of candidate frames. ***All of these methods face limitations, specifically, these methods lack a clear training goal for assessing the quality of selected frames, making it challenging to consistently produce optimal results.***

To address these limitations, we propose a novel learning framework, ViaRL, that incorporates rule-based reinforcement learning (RL) into the video CoT pipeline to optimize frame selection for temporal grounding. ***Unlike previous approach that relies on supervised fine-tuning with pseudo-labels, ViaRL uses the answer accuracy of a downstream MLLM as a reward signal, enabling a trial-and-error learning process that eliminates the need for expensive frame selection annotations***. This approach aligns more closely with human-like learning, where individuals refine their perceptual skills through interaction and feedback rather than exhaustive supervision. By leveraging reinforcement learning, ViaRL dynamically trains a lightweight frame selector to identify the most relevant frames for a given query, ensuring that the model focuses on the key moments that contribute to accurate answer generation.

Inspired by the concept of iterated distillation and amplification introduced in AI 2027 (Kokotajlo et al., 2025), we adopt an iterative training strategy, referred to as Visual Iterated Amplification System, to progressively improve the performance of both the frame selector and the downstream MLLM. They (Kokotajlo et al., 2025; Christiano et al., 2018) decompose a complex problem into multiple simple sub-problems and handle them in parallel, enabling the model to progressively adapt to more challenging tasks. In comparison, we employ a sequential processing approach, which is better aligned with the requirements of our task. Initially, the frame selector is trained using RL to optimize its selection effectiveness, based on the reward signal provided by the downstream MLLM's accuracy. Once the selector achieves a certain level of performance, we freeze it and fine-tune the downstream MLLM to maximize its ability to generate accurate answers using the selected frames. As the downstream model improves, the selector is retrained to further refine its frame selection process, creating a feedback loop where both components collectively enhance their performance. This iterative process ensures that the pipeline adapts to increasingly complex scenarios, enabling robust temporal grounding across diverse video understanding tasks.

By leveraging reinforcement learning and iterative optimization, ViaRL provides a flexible and human-inspired solution to intention-driven video understanding, setting a new exploration for temporal grounding in multimodal tasks. We evaluate ViaRL extensively across multiple benchmarks, including VideoMME (Fu et al., 2024), LVBench (Wang et al., 2024), and MLVU (Zhou et al., 2024),

demonstrating its effectiveness and scalability. Notably, ViaRL achieves significant improvements in temporal grounding performance compared to state-of-the-art baselines. For example, ViaRL achieves a 15% improvement on Needle QA, a subset of MLVU that is widely regarded as one of the most suitable benchmarks for evaluating temporal grounding. Additionally, our experiments demonstrate that ViaRL consistently performs well across diverse visual scenarios and question types, underscoring its broad applicability and robustness.

Our contributions can be summarized as follows:

- We propose ViaRL, the first framework to apply rule-based reinforcement learning to temporal grounding in video understanding tasks. By addressing the challenges posed by training signals, ViaRL establishes a learning paradigm, thinking with videos, that is both flexible and human-like.
- We introduce Visual Iterated Amplification Learning System, an iterative training strategy that progressively improves both frame selection and answer generation through feedback loops.
- Extensive experiments across multiple benchmarks featuring diverse task scenes demonstrate that ViaRL consistently outperforms strong baselines in temporal grounding tasks, highlighting its effectiveness and scalability across a broad range of task scenarios.

## 2 RELATED WORKS

A substantial body of research has focused on MLLMs (Lin et al., 2023; Zhang et al., 2023; Bai et al., 2025; Wang et al., 2025b), frame selection (Hu et al., 2025b; Yu et al., 2024; Tang et al., 2025), and the application of RL to LLMs (Shao et al., 2024; Ahmadian et al., 2024). Our work connects the potentially inefficient learning modes of frame selection methods with reinforcement learning applied to multimodal tasks. Given that existing methods lack a clear training goal for assessing the quality of selected frames in the Video-CoT pipeline, largely due to the subjectivity of the task, we propose ViaRL which delegates the optimization of frame selection to a downstream MLLM. See Appendix A for more detailed discussions of the related work.

## 3 METHODS

In this work, we propose a novel learning pipeline for video temporal grounding, referred to as Visual Iterated Amplification Reinforcement Learning (ViaRL). Our approach involves training a video frame selector that uses natural language communication to identify and convey which frames are relevant to a given query. In Section 3.1, we present an overview of the architecture enabling vision-in-the-loop understanding. In Section 3.2, we detail the rule-based rewards designed to guide the learning process. Finally, in Section 3.3, we elaborate on the Visual Iterated Amplification Reinforcement Learning (ViaRL) framework and its implementation. For the preparation of the training dataset, please refer to Appendix B.

### 3.1 VISION-IN-THE-LOOP ARCHITECTURE

Unlike previous methods (Feng et al., 2025), our architecture is designed to identify the most relevant frames in response to a query text through a language-based QA approach. Similar to existing frame selection methods (Hu et al., 2025b;a; Tang et al., 2025), our architecture consists of two MLLMs. The first MLLM functions as a frame selector, while the second MLLM generates answers by thoroughly analyzing the highly relevant frames.

Previous methods, such as AKS (Tang et al., 2025) and CoS (Hu et al., 2025a), do not account for the temporal relationships between frames in their processes. Besides, Frame-Voyager (Yu et al., 2024) requires retrieving a group of frames from a vast number of combinations, which makes it computationally inefficient. While some approaches, such as those in Hu et al. (2025b), incorporate the temporal dimension, their effectiveness is limited by their reliance on pseudo labels. Additionally, both AKS (Tang et al., 2025) and Hu et al. (2025b) utilize auxiliary selection strategies to avoid selecting redundant or overly similar frames. However, the task of selecting frames is inherently less intuitive compared to enabling a model to engage in natural language-based communication.

Our approach addresses this challenge by enabling the MLLM to directly output the serial numbers of selected video frames. *To achieve this, we first address a fundamental issue: Can the model understand the serial number of each frame?* Unfortunately, existing large models lack this capability. Inspired by the effective approach used in NumPro (Wu et al., 2024), we directly add unique numerical identifiers to the bottom right corner of each video frame, as painted on the candidate frames in Fig. 1. This allows MLLMs to locate events temporally without requiring additional training. While NumPro uses this method to retrieve the start and end moments of events, we extend it to locate specific $N$ frames, achieving frame-level temporal grounding—a task that is significantly more challenging than clip-level grounding.

During the temporal grounding process, we further explore the reasoning capabilities of the model. First, the model analyzes the keywords in the query text to identify meaningful frames. Next, it generates detailed visual descriptions of the relevant frames to provide as much information as possible. Finally, the model outputs a list of frame indices containing $N$ selected frame numbers. As illustrated in Fig 1, the model's reasoning process is highly informative and plays a crucial role in moment grounding.

## 3.2 RULE BASED REWARD MODELING

PPO (Schulman et al., 2017) provides strong stability and reliability but suffers from lower efficiency and higher complexity due to its reliance on a separate critic network. GRPO (Shao et al., 2024) improves efficiency by removing the critic and introduces relative policy optimization, but it has moderate stability and risks reward hacking. REINFORCE++ (Hu, 2025) achieves an optimal balance of efficiency and stability by eliminating the critic while incorporating token-level KL penalties and normalizing advantages across the global batch. It has already been validated in the domain of LLMs, as discussed in Logic-RL (Xie et al., 2025). Given that, *we employ a modified version of REINFORCE++ (Hu, 2025; Xie et al., 2025) as our RL algorithm, with a rule-based reward system serving as the primary training signal to effectively guide policy optimization.*

Through extensive experimentation and careful refinement of our reward design, we developed a robust rule-based reward system comprising four distinct types of rewards: Format Reward, Frame Index Reward, Answer Reward, and Response Length Reward. The system prompt, illustrated in 3.2, is used to guide the selector in retrieving relevant frames. For a detailed description of the system prompt, please refer to the complete version provided in the appendix C.

> **System Prompt**
>
> You are an intelligent chatbot designed for selecting the relevant video frames according to a question. ... Your task is to output $N_{select}$ indices of the frames that can help you answer the question better. ... Your output should follow this format strictly: <think>thinking about keywords and visual appearance here </think><index>target list here </index>...

**Format Reward:** We use regular expression extraction to enforce a structured response format. The selector is required to encapsulate its reasoning process within `<think></think>` tags and provide the target frame index list within `<index></index>` tags. The format score ($S_{format}$) is computed as follows:

$$S_{format} = \begin{cases} 1, & \text{if the response format is correct,} \\ 0, & \text{if the response format is incorrect.} \end{cases} \quad (1)$$

**Frame Index Reward:** This component evaluates the correctness of the frame indices provided in the selector's response. To validate the indices, the model must satisfy the following conditions: the number of indices must be exactly $N$, the indices must fall within the range of valid numerical identifiers, and there must be no repetition. The index score ($S_{index}$) is computed as:

$$S_{index} = \begin{cases} 1, & \text{if all conditions are fully satisfied,} \\ 0, & \text{if any condition is violated.} \end{cases} \quad (2)$$

**Answer Reward:** The third component assesses the correctness of the content in the downstream model's response. After validating the format, the model's answer is compared against the ground

truth to ensure accuracy. The answer score ($S_{answer}$) is computed as:

$$S_{answer} = \begin{cases} 2, & \text{if the answer fully matches the ground truth,} \\ 0, & \text{if the answer is wrong or format/index requirements are not met.} \end{cases} \quad (3)$$

**Response Length Reward:** The final component regulates the length of the content in the model's response. We observe that, without proper incentives, the selector might bypass the reasoning process and directly output the index list. Referred from Video-R1 (Feng et al., 2025), we implement a length control reward to address this issue. This reward encourages the model to provide a detailed reasoning process alongside the index list, ensuring a more comprehensive and structured response. The length score ($S_{length}$) is computed as:

$$S_{length} = \begin{cases} 0.2, & \text{if } l_{\min} \leq \text{length} \leq l_{\max}, \\ 0, & \text{otherwise.} \end{cases} \quad (4)$$

According to the observation of the curve of response length varying with time, we set $l_{min} = 80$ and $l_{max} = 512$.

### 3.3 VISUAL ITERATED AMPLIFICATION REINFORCEMENT LEARNING

During the learning process, there is a critical issue that the downstream answer model can constrain the optimization of the selector. For instance, even when the selector chooses excellent frames, the subsequent model may produce an incorrect answer, which can severely impact the rollout selection during reinforcement learning and cause confusion for the selector. To address this, we propose a novel RL learning paradigm called Visual Iterated Amplification Reinforcement Learning (ViaRL).

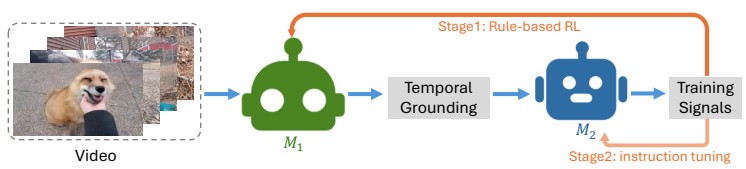

Figure 2: Schematic of our Visual Iterated Amplification System implementation in each cycle.

In this paradigm, as elaborated in Fig 2, training is conducted in alternating phases to optimize both the selector and the answer model effectively. The training signals in the two stages are rewards for RL and labels for next-token prediction, respectively. Initially, the selector undergoes reinforcement learning to achieve strong frame-picking performance. Once the selector demonstrates satisfactory results, we freeze its parameters and switch to instruction tuning of the answer model. As the answer model improves, we unfreeze the selector and retrain it to align with the enhanced performance of the answer model.

During the period of rule-based reinforcement learning (RL) optimization, the policy update is performed using the clipped surrogate objective and defined as follows:

$$\mathcal{J}_{\text{Reinforce++}}(\theta) = \mathbb{E}_{[q \sim P(Q), \{o_i\}_{i=1}^{G} \sim \pi_{\theta_{old}}(O|q)]}$$

$$\frac{1}{G} \sum_{i=1}^{G} \frac{1}{|o_i|} \sum_{t=1}^{|o_i|} \left\{ \min \left[ \frac{\pi_\theta^{i,t}}{\pi_{\theta_{old}}^{i,t}} \hat{A}_{i,t}, \text{clip} \left( \frac{\pi_\theta^{i,t}}{\pi_{\theta_{old}}^{i,t}}, 1-\epsilon, 1+\epsilon \right) \hat{A}_{i,t} \right] \right\}, \quad (5)$$

where:

$$\hat{A}_{i,t} = r(o_{i,<t}) - \beta \cdot \sum_{j=t}^{T} \text{KL}(j), \quad \text{KL}(t) = \log \left( \frac{\pi_{\theta_{old}}^{i,t}}{\pi_{\text{ref}}^{i,t}} \right). \quad (6)$$

Additionally, we normalize this advantage across the global batch for all prompts:

$$\hat{A}_{i,t}^{\text{norm}} = \left\{ \hat{A}_{i,t} - \text{mean} \left( \hat{A}_{i,t} \right) \right\} \Big/ \text{std} \left( \hat{A}_{i,t} \right). \quad (7)$$

The instruction tuning stage follows the general training pipeline outlined in LLaVA (Liu et al., 2023), which is designed to refine the model's ability to understand and respond to natural language

instructions effectively. This stage enhances the model's capability to handle complex queries and adapt to varying frame rates, as the frame rates after frame selection may be arbitrary. This stage ensures that the model can effectively process and reason over selected frames, regardless of their temporal distribution, while maintaining coherence and accuracy in its responses.

Our iterative and alternating training strategy enables mutual refinement between the two components, leading to significant optimization and a more synergistic system. Using a single MLLM for both frame selection and question answering is problematic: if we fine-tune the same model, the answer accuracy from $M_2$, which is crucial for optimizing $M_1$ in Stage 1, may degrade due to the effects of RL fine-tuning, a key distinction from prior works. Moreover, alternately training the models in a single stage would introduce unnecessary complexity into the training pipeline. By keeping the models separate, we not only avoid these issues but also gain flexibility to use a lightweight selector, improving efficiency without compromising overall system performance.

## 4 EXPERIMENTS

### 4.1 SETUP

**Benchmarks.** We conduct experiments on three public benchmarks to evaluate our approach. Video-MME (Fu et al., 2024) comprises 900 videos and 2,700 multiple-choice Question-Answer pairs, categorized into three subsets based on video duration: short (<2 minutes), medium (4∼15 minutes), and long (30∼60 minutes). MLVU (Zhou et al., 2024) includes videos ranging from 3 minutes to 2 hours and spans 9 tasks, with 2,174 multiple-choice VQA pairs. LVBench (Wang et al., 2024) features videos with an average duration of 4,101 seconds per video, which is the longest. It contains 1,549 multiple-choice VQA pairs across 6 tasks. Importantly, all datasets are human-annotated, ensuring high-quality labels for evaluation.

**Training Details.** In this work, we utilize two models with different sizes: Qwen2.5-VL-3B as the selector and Qwen2.5-VL-7B (Bai et al., 2025) as the answer model, since Qwen2.5-VL models could handle videos flexibly with dynamic resolution. As described in Appendix B, we collect 25k pairs for reinforcement learning (RL), out of which 8k pairs are randomly selected as the final RL dataset. For instruction tuning, we randomly select 8k samples from the original LLaVA-Video-178k (Zhang et al., 2024b) dataset. Across different training cycles, the same dataset is reused. All experiments are conducted on 4×A100 80G GPUs, the RL stage takes about 16 hours.

During frame selection, we choose $N$ frames from $T$ candidate frames, with the default configuration being $\{T, N\} = \{128, 8\}$ or $\{256, 16\}$. The resolution of the long side for the two models is resized to $\{112, 896\}$ respectively, while preserving the aspect ratio. This approach ensures that the selector model processes smaller-scale frames for efficient temporal grounding and reasoning during the thinking process.

In order to explore whether the method could generalize to other model, we replace the selector with InternVL3.5-4B (Wang et al., 2025a). $\{T, N\} = \{64, 8\}$, and we use an input resolution of 224×224. Note that InternVL3.5-4B's original resolution is 448×448. Through experiments, we find that InternVL3.5-4B loses basic visual understanding capability when using the 112×112 resolution, whereas 448×448 or 128 frames causes out-of-memory errors.

All experiments are conducted with the selector trained using a constant learning rate of $4.0 \times 10^{-7}$, while the answer model is trained with a learning rate of $1.0 \times 10^{-6}$, batch size is 2 and $G$ in eq.5 is 8. The hyper-parameters for RL are set as follows: $\beta = 1.0 \times 10^{-3}$, $\epsilon = 0.2$.

### 4.2 PERFORMANCE ACROSS GENERAL VIDEO BENCHMARKS

**Temporal Grounding Analysis.** Needle QA is a subset of the MLVU benchmark that requires answering questions related to a specific segment (referred to as the needle) within a longer background video. The dataset is created by randomly inserting the needle into the background video, with a corresponding question-answer pair annotated. This sub-task best reflects the temporal grounding ability of our method. As shown in Table 1, ViaRL(Qwen2.5-VL-3B) achieves a significant improvement, increasing from 58.6 to 73.5 (8 frames), which is nearly a 15% enhancement. Our model achieves the

Table 1: Experimental results on VideoMME (without subtitle assistance), LVBench and MLVU benchmarks. We assess the performance of ViaRL after it undergoes two cycles of learning. (Note: The bold number represents higher number between Qwen2.5-VL and Qwen2.5-VL+ViaRL. Considering that we provide a method which helps existing models further explore their capabilities and improve their performance like a plug-and-play manner, rather than training a new SOTA model. Therefore, it's more important to compare the method with the basic model to demonstrate the effectiveness.)

| Models | Size | frames | VideoMME w/o sub. | | | | LVBench val | MLVU Dev | |
| --- | --- | --- | --- | --- | --- | --- | --- | --- | --- |
| | | | Short | Medium | Long | Avg | Avg | Needle QA | M-Avg |
| Proprietary Models | | | | | | | | | |
| GPT-4o (OpenAI, 2024) | - | 384 | 80.0 | 70.3 | 65.3 | 71.9 | 30.8 | 64.8 | 64.6 |
| Gemini-1.5-Pro (Team et al., 2023) | - | 0.5 fps | 81.7 | 74.3 | 67.4 | 75.0 | 33.1 | - | - |
| Open-source MLLMs | | | | | | | | | |
| MovieChat (Song et al., 2024) | 7B | 2048 | - | - | - | - | 22.5 | 24.2 | 25.8 |
| TimeChat (Ren et al., 2024) | 7B | 96 | - | - | - | - | 22.3 | 24.5 | 30.9 |
| VideoChat2 (Li et al., 2024b) | 7B | 16 | 48.3 | 37.0 | 33.2 | 39.5 | - | - | 44.5 |
| VideoLLaVA (Lin et al., 2023) | 7B | 8 | 45.3 | 38.0 | 36.2 | 39.9 | - | - | 47.3 |
| Sharegpt4Video (Chen et al., 2024a) | 7B | 16 | 48.3 | 36.3 | 35.0 | 39.9 | - | - | 46.4 |
| InternVL-V1.5 (Chen et al., 2024b) | 20B | 10 | 60.2 | 46.4 | 45.6 | 50.7 | | - | 50.4 |
| Video-CCAM (Fei et al., 2024) | 14B | 96 | 62.2 | 50.6 | 46.7 | 53.2 | - | 73.2 | 63.1 |
| LongVA (Zhang et al., 2024a) | 7B | 128 | 61.1 | 50.4 | 46.2 | 52.6 | - | 69.3 | 56.3 |
| Video-XL (Shu et al., 2024) | 7B | 128 | 64.0 | 53.2 | 49.2 | 55.5 | - | 73.8 | 64.9 |
| Kangaroo (Liu et al., 2024) | 8B | 64 | 66.1 | 55.3 | 46.7 | 56.0 | - | - | - |
| Qwen2.5-VL (Bai et al., 2025) | 7B | 8 | 61.7 | 50.6 | 46.3 | 52.9 | 32.3 | 58.6 | 54.5 |
| +ViaRL(Qwen2.5-VL-3B) | 7B | 8 | 65.1 | **56.1** | **50.8** | 57.3 | 36.9 | 73.5 | 58.2 |
| +ViaRL(InternVL3.5-4B (Wang et al., 2025a)) | 7B | 8 | **66.9** | 55.9 | 50.0 | **57.6** | **38.2** | **76.3** | **58.4** |
| Qwen2.5-VL | 7B | 16 | 67.6 | 57.0 | 49.0 | 57.9 | 34.9 | 60.6 | 55.7 |
| +ViaRL(Qwen2.5-VL-3B) | 7B | 16 | **68.1** | **57.4** | **52.8** | **59.4** | **37.7** | **76.1** | **61.1** |

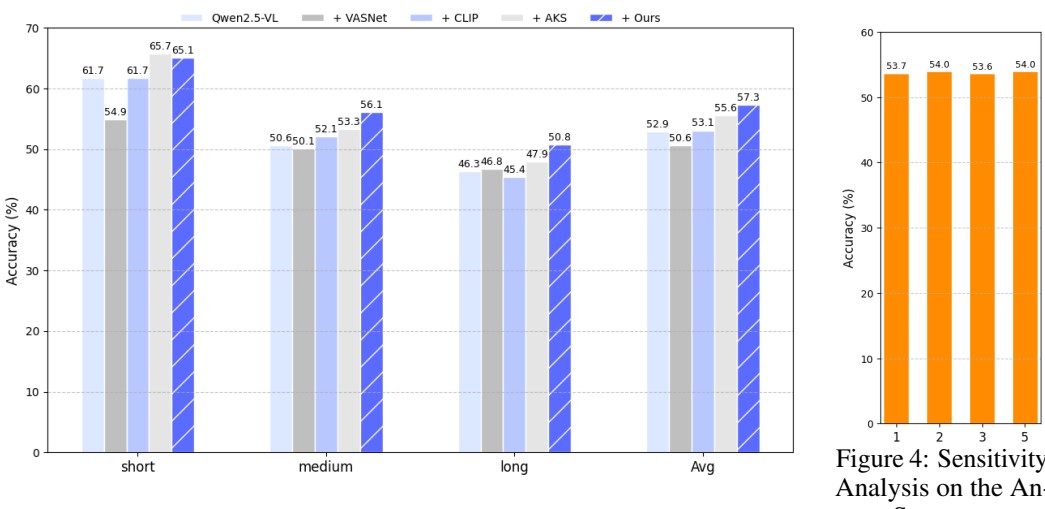

Figure 3: Comparison of Frame Selection Models.

Figure 4: Sensitivity Analysis on the Answer Score.

performance of Video-CCAM with 96 frames and Video-XL with 128 frames using only 8 frames. Improvements are also observed at 16 frames.

**Comparison of Frame Selection Models.** As shown in the Fig 3, we compare the basic Qwen2.5-VL with different frame selection models on VideoMME with 8 frames, including VASNet (Fajtl et al., 2019), CLIP (Radford et al., 2021), AKS (Tang et al., 2025), and ViaRL. Compared with the recent frame selection method AKS, our ViaRL achieves consistently better overall performance. In particular, the improvements on the long-video subset are the most indicative: they suggest that ViaRL is more capable of selecting informative and contextually relevant frames. This observation further supports our claim that the proposed RL–based policy is more effective than heuristic selection strategies such as AKS.

**Quantitative Analysis.** As shown in the Table 1, ViaRL brings consistent accuracy gain over three long video understanding benchmarks. In the VideoMME benchmark, the Qwen2.5-VL+ViaRL(Qwen2.5-VL-3B) with 8 frames achieves an average score of 65.1 for short videos, 56.1 for medium, 50.8 for long, and an overall average of 57.3. Compared to basic Qwen2.5-VL,

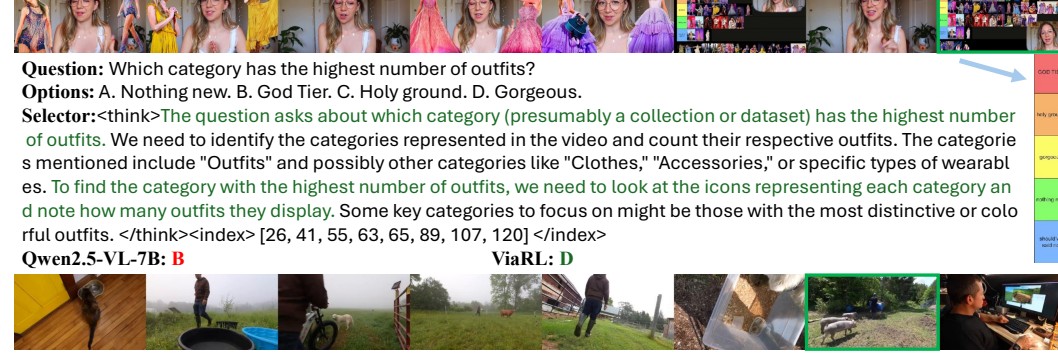

**Question:** Which category has the highest number of outfits?
**Options:** A. Nothing new. B. God Tier. C. Holy ground. D. Gorgeous.
**Selector:**<think>The question asks about which category (presumably a collection or dataset) has the highest number of outfits. We need to identify the categories represented in the video and count their respective outfits. The categories mentioned include "Outfits" and possibly other categories like "Clothes," "Accessories," or specific types of wearables. To find the category with the highest number of outfits, we need to look at the icons representing each category and note how many outfits they display. Some key categories to focus on might be those with the most distinctive or colorful outfits. </think><index> [26, 41, 55, 63, 65, 89, 107, 120] </index>
**Qwen2.5-VL-7B: B**                    **ViaRL: D**

**Question:** What is the purpose of the man in the video processing the blue plastic barrel?
**Options:** A. Processing the blue plastic barrel into troughs for feeding cattle. B. Processing the blue plastic barrel into troughs for feeding ducks. C. Processing the blue plastic barrel into troughs for feeding chicken. D. Processing the blue plastic barrel into troughs for feeding pigs.
**Selector:**<think>The keywords focus on a man working on a blue plastic barrel. This likely includes activities such as sorting, packaging, or processing of materials. The frames need to show the man interacting with the object and the surrounding environment. Since the work seems to be done outdoors in an agricultural setting, frames depicting grassy areas, machinery, and possibly warehouse environments would be crucial. </think><index> [5, 18, 32, 44, 56, 70, 86, 99] </index>
**Qwen2.5-VL-7B: B**                    **ViaRL: D**

Figure 5: ViaRL improves the baseline MLLMs for video understanding. The $N$ selected frames are shown. The most relevant frame is indicated by green box in each row.

the method equipped with ViaRL model shows a marked improvement across all categories. In the LVBench benchmark, Qwen2.5-VL+ViaRL(Qwen2.5-VL-3B) achieves a validation score of 36.9, outperforming other commercial models like GPT-4o and Gemini-1.5-Pro. This highlights its effectiveness in video content analysis tasks with limited frame data. In the MLVU benchmark, the M-Avg score is 58.2, further showcasing its robust performance across different evaluation tasks. Furthermore, *we observe that the performance of Qwen2.5-VL with 16 frames is comparable to Qwen2.5-VL+ViaRL with only 8 frames*. Overall, the integration of ViaRL's cycle-based optimization strategy significantly enhances the video content processing capabilities of the Qwen2.5-VL+ViaRL model, making it a strong competitor against other models in scenarios with limited frames.

**Qualitative Analysis.** The Fig 5 showcases the visualization of frame selection process in our ViaRL, which involving visual comprehension and analysis associated with a question. We can see from the comparison of ViaRL and the baseline, ViaRL performs better by useful temporal grounding drawn by the green box. More visualization results are displayed in Appendix F.

**Inference Cost.** Given that inference time is directly dependent on hardware performance, we use relative time for evaluation in order to mitigate the influence of the machine itself. And it is estimated using one NVIDIA A100-80G GPU with one sample. Specifically, in the test of selecting 8 frames from 128 on Video-MME, we use the time relative to directly performing question answering on 128 frames with the same resolution in $M_2$ as the metric. Our method takes 75% of the time required to process all frames, with the selector accounting for 57% and the answer model for 18%. It indicates that this method is more efficient than processing all frames.

### 4.3 GENERALIZATION

**Generalization to other models.** As shown in Table 1, we replace the selector model from Qwen2.5-VL-3B with InternVL3.5-4B. Equipped with InternVL3.5-4B, the base model achieves improvements across all benchmarks. On the Needle-QA, it obtains a gain of 17.7. Besides, as shown in Table 4, we have also tested the MiMo-VL-7B, which acts as a selector, and obtained a similar improvement. Both MiMo-VL-7B and InternVL3.5-4B show improvements with our method, demonstrating its ability to generalize to other models.

**Generalization to other tasks.** Our proposed ViaRL performs well not only on multiple-choice VQA benchmarks, but also on more general tasks such as open-ended answer generation.

Table 2: Experimental results on Video-based Text Generation Performance Benchmark.

| Model | Correctness of Information | Detail Orientation | Contextual Understanding |
|---|---|---|---|
| Qwen2.5-VL-7B | 2.43 | 2.58 | 3.06 |
| +ViaRL(Qwen2.5-VL-3B) | 2.51 | 2.62 | 3.10 |

We randomly select $8k$ open-ended VQA samples from the LLaVA-Video-178K dataset and perform a single-step optimization of the selector(Cycle1-Stage1). The selector identifies 8 frames from the 128-frame input. We then evaluate the model using the Video-based Text Generation Performance Benchmark introduced in Video-ChatGPT (Maaz et al., 2024), which has 1996 open-ended QA pairs.

The results are displayed in Table 2. We follow the evaluation setting of Video-ChatGPT, using GPT-3.5 and scoring $0 \sim 5$ to assess the response quality across several dimensions, such as Correctness of Information, Detail Orientation and Contextual Understanding. These results demonstrate that ViaRL continues to perform effectively on the open-ended QA task.

## 4.4 ABLATION STUDY

**Sensitivity Analysis on the Answer Score.** In the Sec 3.2, we introduce 4 different rewards. Answer is the most crucial reward for optimizing the selector, and training is impossible without it. Without the frame index or format reward, it is difficult to obtain $N$ distinct frame indices in the correct format from the selector's output throughout training, and as a result, the answer model cannot receive valid video frame inputs. It would make the training ineffective.

The answer reward plays the most critical role. To assess its impact, we conduct experiments by varying the weight of the answer reward (set to 1, 2, 3, and 5, respectively) while keeping the other rewards fixed at Cycle1-Stage1. As shown in the Fig 4, ViaRL is not sensitive to the choice of answer reward weights. The performance metrics remain relatively stable across different values, with only minor fluctuations observed.

**Different Cycles and Stages.** Across these benchmarks, there is a consistent trend of performance improvement with each cycle and stage on the whole, highlighting the effectiveness of the iterative learning strategy employed by the ViaRL model (8 frames), as displayed in Fig 6. Notably, the performance of VideoMME-long and Needle QA on the MLVU dataset improves significantly when transitioning from cycle-stage pair $(1, 2)$ to $(2, 1)$, which corresponds to RL learning after completing one cycle.

As the model progresses through cycles, the rate of improvement begins to taper, indicating diminishing returns with additional cycles. After all, there is currently no perfect multimodal large model capable of providing ideal answers based on the selected video frames all the time. We use the selector from Cycle2–Stage1 and replace $M_2$ with Qwen2.5-VL-32B as the answer model, which provides better answer reward. All other settings remain unchanged and we then conduct RL for

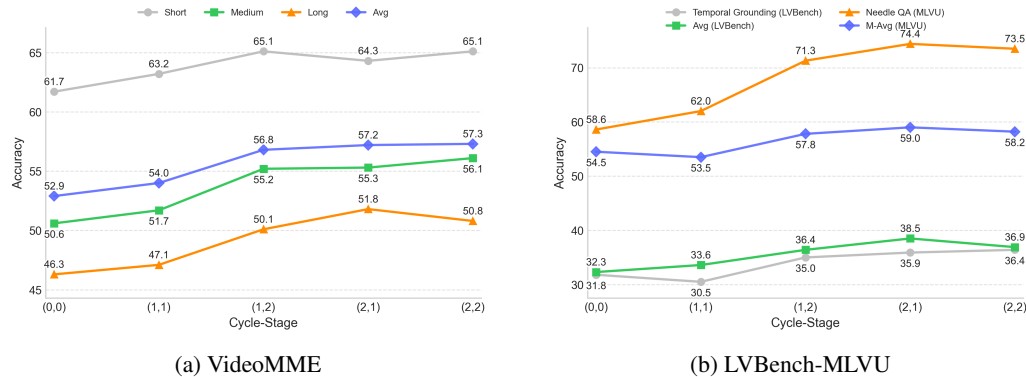

(a) VideoMME      (b) LVBench-MLVU

Figure 6: Performance of our ViaRL over multiple cycles and stages, attributing to the intertwined improvement of models capability during the iterative process. The horizontal axis $(i, j)$ represents the $j_{th}$ stage of the $i_{th}$ cycle. For example, $(2, 1)$ indicates the evaluation model $M_1$ has learned twice, and $M_2$ has learned once. The initial state is denoted as $(0, 0)$.

Table 3: VideoMME w/o sub. performance metrics for different training recipes (8 frames). All RL evaluations are conducted after the $1_{th}$ stage of the $1_{th}$ cycle.

| Method | w/ SFT | w/ think | w/ length reward | Data Num | short | medium | long | Avg |
|--------|--------|----------|------------------|----------|-------|--------|------|-----|
| baseline | - | - | - | - | 61.7 | 50.6 | 46.3 | 52.9 |
| SFT | ✓ | ✗ | ✗ | 30k | 59.9 | 49.6 | 46.2 | 51.9 |
| RL | ✓ | ✓ | ✓ | 30k+8k | 58.9 | 47.8 | 44.8 | 50.5 |
| RL | ✗ | ✗ | ✓ | 8k | 58.2 | 51.3 | 45.9 | 51.8 |
| RL | ✗ | ✓ | ✗ | 8k | 59.1 | 50.7 | **47.3** | 52.4 |
| RL | ✗ | ✓ | ✓ | 8k | **63.2** | **51.7** | 47.1 | **54.0** |

the selector, the average performance on VideoMME increases from 56.7 to 57.6. It indicates the selector remains improvable when paired with a stronger answer model. Furthermore, the capabilities of using only 8 frames are inherently limited. Therefore, these limitations don't prevent us from concluding the effectiveness of multi-cycle training.

**Different Training Recipes.** The results are listed in Table 3. Given that the initial model exhibits poor frame-level temporal grounding capacity with text queries, we incorporate the model fine-tuned through SFT as a starting point for RL. As outlined in Appendix B, we construct the SFT dataset and the selected $N$ frames are relevant and can serve as pseudo labels for the SFT process.

We have collected 30k SFT data points, each paired with pseudo-labels specifying $N$ relevant frame indices. Leveraging this dataset, we use SFT to train a model capable of directly predicting the relevant frames. Subsequently, we use this SFT-trained model as the initial policy for reinforcement learning (RL), leveraging its existing ability for frame grounding to some extent. However, the results deteriorate, with the average accuracy decreasing from 51.9% to 50.5%. *Based on our observations, this decline is primarily due to the SFT-trained model inheriting the shortcomings of the CLIP model. Therefore, the SFT-trained model is not an ideal starting point for RL.*

By leveraging the training signals provided by the answer model, the "RL without thinking" approach can also be trained. However, its average accuracy is 51.8%, and the average accuracy without applying the length reward is 52.4%. Both are noticeably lower compared to our method, which achieves an average accuracy of 54.0%. These results indicate that our approach, which incorporates a rich and meaningful thinking process, significantly enhances the model's ability to select relevant frames more effectively.

### 4.5 SCALING BEHAVIOR

Table 4: Performance when scaling model parameters.

| Models | short | medium | long | Avg |
|--------|-------|--------|------|-----|
| Qwen2.5-VL($M_2$) | 61.7 | 50.6 | 46.3 | 52.9 |
| +Qwen2.5-VL-3B | 63.2 | 51.7 | 47.1 | 54.0 |
| +MiMo-VL-7B | **63.8** | **52.8** | 47.2 | 54.6 |
| +Qwen2.5-VL-7B | 62.7 | 52.3 | **49.8** | **54.9** |

We set up different sizes of selector, respectively 3B and 7B (more larger models are beyond our computation resource). We adopt MiMo-VL-7B (Team et al., 2025) or Qwen2.5-VL-7B (Bai et al., 2025) due to their adaptability for varying resolution. As reported in Table 4, results are evaluated on VideoMME with 8 frames at Cycle1-Stage1, and our method performs better with larger selector.

### 5 CONCLUSION

In this work, we proposed ViaRL, a novel framework that integrates rule-based RL into the video CoT pipeline to address the challenges of temporal grounding in multimodal video understanding task. By delegating the optimization of frame selection to a downstream MLLM and leveraging a reward-driven trial-and-error learning process inspired by human-like perceptual refinement, ViaRL eliminates the lacking of frame selection annotations and dynamically trains a lightweight frame selector to focus on the most relevant frames for further accurate answer generation. Through an iterative optimization strategy, referred to as the Visual Iterated Amplification Learning System, ViaRL progressively enhances the performance of both the frame selector and downstream multimodal large language models, adapting to increasingly complex scenarios and ensuring robust temporal grounding.

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

## A    MORE RELATED WORKS

**MLLMs for Video Understanding.** Recent advancements in Multimodal Large Language Models (MLLMs) have transformed video understanding through unified representation learning, efficient temporal modeling, and scalable architectures. Video-LLaVA (Lin et al., 2023) pioneered alignment of image and video features into a shared language space, achieving state-of-the-art performance on video QA benchmarks. Video-LLaMA (Zhang et al., 2023) extended this by integrating audio-visual cues via modality-specific Q-formers. Scalability challenges in high-resolution and long videos were addressed by Qwen2.5-VL (Bai et al., 2025), which introduced dynamic FPS sampling for video processing. Long-context modeling saw innovations like Video-XL (Shu et al., 2024), compressing hour-long videos hierarchically, and LongViTU (Wu et al., 2025), emphasizing long video context and condensed reasoning. LLaVA-NeXT-Interleave (Li et al., 2024a) unifying multi-image, video, and 3D tasks. InternVideo2.5 (Wang et al., 2025b) developed compact spatiotemporal representations through adaptive hierarchical token compression. Despite the progress, the approach still differs from the way humans process information, as it relies on uniform sampling employed by these models.

**Frame Selection.** Efficient frame selection has become pivotal for scalable long-video understanding, evolving from traditional redundancy-reduction approaches like uniform sampling or clustering-based methods to modern query-adaptive strategies. Early methods such as Video Summarization (Fajtl et al., 2019; Son et al., 2024) focused on generic keyframe or keyshot extraction but lacked task-specific alignment with text, while contemporary techniques leverage multimodal large language models (MLLMs) for dynamic adaptation. M-LLM Based Frame Selection (Hu et al., 2025b) employs spatial-temporal importance scoring to boost performance, and Frame-Voyager (Yu et al., 2024) ranks frame combinations via pre-trained Video-LLMs. Adaptive Keyframe Sampling (AKS) (Tang et al., 2025) jointly maximize prompt relevance and frame coverage through lightweight modules. Complementary methods include Chain-of-Shot (CoS) (Hu et al., 2025a) exploring MLLMs' summary capacity for binary coding and pseudo temporal grounding on long videos.

**Reinforcement Learning.** Recent progress in RL has emphasized stable, efficient, and interpretable policy optimization. Trust Region Policy Optimization (TRPO) (Schulman et al., 2015) introduced trust region constraints via KL divergence to ensure monotonic policy improvement, avoiding catastrophic updates in neural network training. Proximal Policy Optimization (PPO) (Schulman et al., 2017) simplified TRPO's constraints by replacing them with a clipped objective function, enabling stable first-order optimization with lower computational costs. Further innovations like Group Relative Policy Optimization (GRPO) (Shao et al., 2024) eliminated value networks in favor of group-wise KL penalties, reducing memory usage in language model alignment while maintaining training stability. Reinforce++ (Hu, 2025) combined REINFORCE's simplicity with PPO-like stability mechanisms, removing critic networks to reduce complexity. REINFORCE Leave-One-Out (RLOO) (Ahmadian et al., 2024) minimized gradient variance through leave-one-out estimation, outperforming PPO in multilingual tasks. Efficiency-focused methods like ReMax (Li et al., 2023) accelerated training for large language models via greedy baselines. Collectively, these methods bring innovations from robotic control to language alignment, emphasizing sample efficiency and stability in complex reasoning scenarios.

## B    DATASET PREPARATION

We utilize a subset of the LLaVA-Video-178K (Zhang et al., 2024b) dataset and perform a filtering operation. First, the CLIP-ViT-Large (Radford et al., 2021) model is employed to select $N$ frames based on their top-$N$ cosine similarity to the question text. Next, a MLLM is used to predict answers based on these $N$ selected frames. Additionally, predictions made without incorporating visual information are considered. Let the question be denoted as $Q$, the selected frames as $F_s$, and the correct answer as $GT$. The prediction without using visual information is denoted as $Pred_1 = MLLM(Q)$, while the prediction that incorporates the selected frames is denoted as $Pred_2 = MLLM(Q, F_s)$.

For each question-answer pair, we filter out cases where the correct prediction is made without video input, as these may have been guessed correctly. This corresponds to cases where: $GT \neq Pred_1$. Next, we gather cases where the prediction is incorrect with visual information used, which satisfies: $GT \neq Pred_2$. These cases represent challenging examples that can be utilized for Reinforcement

Learning (RL). Sometimes, a frame could be barely connected with the question when it is alone but important when in frames context. However, such case would not appear or their importance would be overshadowed in samples where $GT == Pred_2$ because useful visual info is very likely already contained in the CLIP-selected relevant frames. Conversely, This case is more likely to be found in samples where $GT \neq Pred_2$. Besides, we construct the SFT dataset by collecting cases where the correct answer is predicted using the selected frames, satisfying the conditions: $GT \neq Pred_1$ and $GT = Pred_2$. In our implementation, Qwen2.5-VL-3B (Bai et al., 2025) is used as the MLLM in this phrase.

## C    DETAILS OF SYSTEM PROMPT

---

**System Prompt**

You are an intelligent chatbot designed for selecting the relevant video frames according to a question.

User will provide you a video with $N_{candidate}$ frames and a short question.

The red numbers in the bottom right corner of each frame represent the frame indice. The frame index is an integer in the range of 0 to $N_{candidate} - 1$.

Your task is to output $N_{select}$ indices of the frames that can help you answer the question better.

Here's how you can accomplish the task:

1. Think about the keywords from the question:
- Check if the physical entities are mentioned.
- Check if the occurrence time is mentioned.
- Check if the place or location is mentioned.
- Check if the action is mentioned.
2. Provide the appearance reference based on the keywords and video:
- Describe the visual appearance of the $N_{select}$ frames that are most relevant to the keywords.
3. Provide the target list:
- A list of $N_{select}$ frame indices, that the corresponding frames are most helpful to answer the question.

Your output should follow this format strictly:

<think>thinking about keywords and visual appearance here </think><index>target list here </index>.

Specific requirements are as follows:

**Ensure that anyone can uniquely identify these target frames in the video through the references.**

**Ensure that the references are complete and independent.**

**Don't output the words '<think>thinking about keywords and visual appearance here </think>' directly.**

**Ensure that the list consists of $N_{select}$ values.**

---

## D    FURTHER ELABORATION ON NOVELTY.

Introducing frame selection before question answering is not the distinctive aspect in our paper.

As we all know, o3 provides a paradigm for reasoning within images, maintaining visual information throughout the process and focusing on relevant regions. Inspired by it, we explore how to identify the most relevant frames to a given query in a video sequence.

While there are several frame-selection methods (Tang et al., 2025; Hu et al., 2025b;a), all of them lack a clear training goal for assessing the quality of selected frames in the Video-CoT pipeline, primarily because the task is inherently subjective. Consequently, these methods can't be optimized through existing GRPO-based techniques.

Therefore, we propose ViaRL which delegates the optimization of frame selection to a downstream MLLM. Besides, ViaRL uses the answer accuracy as a reward signal. This enables a trial-and-error

learning process, eliminating the need for expensive frame selection annotations like pseudo labels or human-annotated labels.

In fact, this method for optimizing selected frame quality is proposed for the first time.

## E  THE USE OF LARGE LANGUAGE MODELS

In response to the question "Did you use Large Language Models (LLMs) in paper writing?", we clarify that LLMs were only used to polish the writing.

## F  MORE VISUALIZATION RESULTS

In Figures 7, 8, and 9, we present additional examples of video understanding results achieved by our ViaRL. These examples demonstrate the consistent performance of ViaRL across a wide range of visual scenarios and question types, highlighting its versatility and robustness in diverse applications.

**Question:** Based on the video, which of the following describes the reason why the student ate the banana?

**Options:** A. Because the banana looks tasty. B. Because he considered the process of eating a banana is art. C. Because he didn't think the banana worth $120,000. D. Because he wanted to followed the man who ate a banana in a exhibition in 2019.

**Selector:**<think>The keywords 'banana' and 'students' are present in the question, so I am looking for frames where the student is seen eating or holding the banana. This might involve finding frames showing the student, such as in close-up shots or with their hands in the foreground. Additionally, the frames should have clear visual cues of the banana like its color, texture, or the hands handling it, so they stand out clearly against the background. Also, looking for the student in different environments like classrooms or corridors since the question suggests location. Lastly, paying attention to transitions and possibly highlighting the frames that show the student interacting with the banana or its surroundings.</think><index> 5, 11, 17, 24, 30, 36, 42, 48 </index>

**Qwen2.5-VL-7B: C**                        **ViaRL: B**

**(1) Action Reasoning**

**Question:** Where does mRNA come from in the video?

**Options:** A. Cells. B. Vaccines. C. COVID-19 virus. D. Antibodies.

**Selector:**<think> The question asks about the origin of mRNA in the video. Keywords such as "mRNA" and "where" indicate we need to focus on parts of the video where the source of mRNA is shown or described. We should look at scenes involving mRNA synthesis, cellular activities, and possibly interactions between mRNA and other components in the cell.</think><index> 3, 49, 65, 25, 6, 95, 73, 33 </index>

**Qwen2.5-VL-7B: C**                        **ViaRL: B**

**(2) Attribute Perception**

**Question:** How many individuals are visible in the introductory shot of the video?

**Options:** A. 5. B. 6. C. 7. D. 4.

**Selector:**<think>In the introductory shot, there is a man in front of a podium speaking to an audience. To determine the number of individuals, I'll consider objects outside the main speaker. At least five other people are visible in the frames before the intro begins. Two are seated at the front of the room, one to the right side, another further back, and one last person partially cut off on the left side. <think><index>4, 5, 6, 7, 11, 19, 25, 34</index>

**Qwen2.5-VL-7B: D**                        **ViaRL: A**

**(3) Counting Problem**

**Question:** What does the man use to clean the keyboard in this video?

**Options:** A. The man uses a non-abrasive sponge soaked in water to clean the keyboard. B. The man uses a soft-bristled brush to scrub the keyboard keys. C. The man uses a microfiber cloth to wipe off the keyboard. D. The man uses dry compressed air to remove dust from the keyboard.

**Selector:**<think>To address the question, we need to focus on identifying relevant frames where cleaning activities occur. These could be frames showing the keyboard, a cleaning substance being used, or the man using a cleaning tool. The sequence starting from frame 6 shows multiple screen elements like icons on the desktop and windows which might indicate some interaction unrelated to cleaning. However, the last visible frame before our search threshold has the man visibly interacting with the keyboard.</think><index>6, 56, 40, 30, 28, 20, 12, 8</index>

**Qwen2.5-VL-7B: B**                        **ViaRL: D**

**(4) Object Recognition**

**Question:** According to the video, which of the following is considered the earliest stage of human evolution?

**Options:** A. Ramapithecus. B. Dryopithecus. C. Ardipithecus Ramidus. D. Homo Sapiens.

**Selector:**<think> The keywords from the question involve human evolution stages. The question is asking about the earliest stage of human evolution within the given timeline of frames. The appearance of each frame shows different stages of early human evolution, so it's important to highlight the ones where humans start to walk upright and show bipedalism. This will help demonstrate the advancement from ape-like to fully human forms. </think><index> 0, 4, 16, 20, 32, 44, 50, 127 </index>

**Qwen2.5-VL-7B: A**                        **ViaRL: B**

**(5) Temporal Perception**

Figure 7: Visualization across diverse scenarios on VideoMME.

**Question:** What does the man do after he sees the woman for the third time?
**Options:** A. He runs a marathon. B. He sleeps in his bed. C. He draws the woman in the book. D. He looks for the woman.
**Selector:**<think> The question asks about what the man does after seeing the woman for the third time. It implies a sequence where the man initially sees the woman less often, then sees her more frequently, culminating in at least three meetings. I need to find frames where the interaction between the characters reaches this milestone. </think><index> 9, 15, 18, 29, 35, 36, 43, 49 </index>
**Qwen2.5-VL-7B: D**                            **ViaRL: C**
**(1) Event Understanding**

**Question:** What color tie is the host wearing?
**Options:** A. Purple. B. Red. C. White. D. Black.
**Selector:**<think> The question asks for the color of the host's tie, which would typically be one of the key features in a video identifying a person. Since the name "Kenan" is part of the description and he is giving a speech, it might be inferred that the question refers to Kenan Thompson, who has a distinctive tie color. </think><index> {4, 6, 9, 15, 28, 35, 46, 76} </index>
**Qwen2.5-VL-7B: D**                            **ViaRL: A**
**(2) Key Information Retrieval**

**Question:** What is Kobe's sitting posture at 05:32?
**Options:** A. Sitting upright. B. Sitting with legs crossed. C. Sitting leaning forward. D. Legs together.
**Selector:**<think> The question asks about Kobe's sitting posture at a specific time during a video, likely showing a sequence where Kobe is seated and possibly speaking or reacting to something. Given that the frame indices need to be chosen based on the timestamp, identifying moments where Kobe is visible in a seated position at or around 5:32 is crucial. Considering typical sitting postures and sequences in such media, looking for frames around these timestamps would help highlight when Kobe might be seated correctly for posture analysis. </think><index> 42, 56, 64, 77, 90, 106, 118, 124 </index>
**Qwen2.5-VL-7B: A**                            **ViaRL: B**
**(3) Temporal Grounding**

**Question:** Why does the vlogger do push-ups in hotel?
**Options:** A. He is in a push-up competition. B. Hotel asks all guests to do so. C. He eats too much food these days. D. He is engaged in a workout.
**Selector:**<think> The question asks about the reason behind the vlogger doing push-ups in a hotel. Key elements to consider include the vlogger's activities, the setting (the hotel room), and any visible actions (push-ups). To find frames that are most relevant, focus on scenes showing a person exercising or performing physical activity in a hotel-like setting. </think><index> 8, 9, 10, 35, 52, 64, 89, 116 </index>
**Qwen2.5-VL-7B: D**                            **ViaRL: C**
**(4) Reasoning**

**Question:** What are the key events or actions that occur from 01:17 - 05:07?
**Options:** A. The photographer has recorded the magnificent marine landscape. While seated in a pneumatic boat and diving, he captured scenes of pods of sperm whales active in the area. B. The photographer captured scenes of pods of orca active in the marine area from aboard a cruise ship. C. Many sperm whales are active in this sea region, and the photographer, seated in a helicopter, pursues them to film and document their behavior. D. The documentary captures the terrain and weather conditions of the filming location, where humans are fishing in the marine area. Numerous orca are active in this sea region, and the photographer, seated in a pneumatic boat, dives to pursue and film their behavior.
**Selector:**<<think> The event being described likely involves a sequence of actions or changes in the scene from approximately 01:17 to 05:07 in the video. This could involve a transition from one scene to another, such as a change in weather, time, or a new character entering the scene. The goal is to pick frames that depict significant changes or actions at these timestamps. </think><index> [47, 52, 58, 60, 76, 80, 107, 120] </index>
**Qwen2.5-VL-7B: C**                            **ViaRL: D**
**(5) Summarization**

Figure 8: Visualization across diverse scenarios on LVBench.

**Question:** What is the main scene in the video?
**Options:** A. Sky. B. Barren land. C. Ocean. D. Wetland.
**Selector:**<think>The question asks about the main scene in the video, and it likely focuses on the context and environment of the footage. Key visual elements such as elephants, text references, and background descriptions should be considered.</think><index>5, 10, 25, 42, 52, 69, 89, 118</index>
**Qwen2.5-VL-7B: D**                    **ViaRL: B**
**(1) Topic Reasoning**

**Question:** Does this surveillance footage contain any anomalies? If yes, which kind of anomaly?
**Options:** A. Vandalism. B. Abuse. C. Assault. D. Burglary.
**Selector:**<think> The question asks about anomalies in the surveillance footage. Anomalies could include unusual behavior of the people present, anything not behaving as expected, such as sudden movements, interactions away from normal activity, or anything uncharacteristic. I'll focus on frames where actions or behaviors appear non-normal.
</think><index> [0, 4, 8, 9, 19, 29, 55, 74] </index>
**Qwen2.5-VL-7B: C**                    **ViaRL: B**
**(2) Anomaly Recognition**

**Question:** What are the two young women wearing in the street?
**Options:** A. Trendy summer clothes. B. Swimwear. C. Formal attire. D. Winter clothes.
**Selector:**<think> The question specifies that we need to find the two young women in the street and their clothing. The video shows various scenes where the key characters interact, so I need to look for moments where two women are walking together outdoors. Given that they are described as young and women, it's likely they are carrying bags which further suggests they might be talking while walking. </think><index> 45, 60, 83, 94, 98, 112, 116, 123 </index>
**Qwen2.5-VL-7B: C**                    **ViaRL: A**
**(3) Needle QA**

**Question:** Where did I put the blue helmet?
**Options:** A. on the wall hanger. B. on the kitchen counter. C. under the bed. D. in the closet.
**Selector:**<think> The question asks about the location of a blue helmet, which suggests two things - the presence of a blue helmet and its lost or misplaced state. In the video, there is a scene where a person puts a helmet on a counter near a bookcase, indicating it might be in close proximity to other objects like books or a door, usually found near entryways or hallways rather than directly on the floor. </think><index> 5, 49, 62, 65, 106, 108, 124, 126 </index>
**Qwen2.5-VL-7B: B**                    **ViaRL: D**
**(4) Ego Reasoning**

**Question:** What did the cartoon mouse do to the cartoon cat's tail?
**Options:** A. Cut it off. B. Trimmed the fur. C. Used it as a candle. D. Nailed it.
**Selector:**<think> The question is about what the mouse did to the cat's tail. It's important to find instances where the mouse is seen harming or interacting closely with the cat's tail, such as biting it or grabbing it. Key visuals include moments where the mouse has contact with the cat's body or its tail. </think><index> 8, 14, 18, 22, 26, 56, 62, 116 </index>
**Qwen2.5-VL-7B: A**                    **ViaRL: C**
**(5) Plot QA**

Figure 9: Visualization across diverse scenarios on MLVU.

