# OpenReview forum: "ViaRL: Adaptive Temporal Grounding via Visual Iterated Amplification Reinforcement Learning"
_ICLR.cc/2026/Conference — Submitted to ICLR 2026_

### Official Review · Reviewer_4Vp9 · 2025-10-24

**Soundness:** 3
**Presentation:** 3
**Contribution:** 4
**Rating:** 8
**Confidence:** 3

**Summary:**

The paper addresses the challenge of intention-driven video understanding, focusing on selecting relevant frames in videos to answer queries effectively. The authors propose ViaRL, a novel framework that uses rule-based RL to optimize frame selection by leveraging the answer accuracy of a downstream MLLM as a reward signal. This trial-and-error learning approach eliminates the need for expensive annotations and mimics human-like learning.

**Strengths:**

The paper presents a novel framework, ViaRL, which leverages rule-based reinforcement learning to optimize frame selection in video understanding tasks. Central to the approach is the Visual Iterated Amplification training strategy, an innovative iterative refinement process that alternates between optimizing the frame selector and the answer model, providing strong motivation and technical soundness. The effectiveness of ViaRL is demonstrated through comprehensive experiments on several challenging benchmarks, including VideoMME, LVBench, and MLVU, where it consistently achieves improvements. Given the difficulty of the tasks and the good results, I believe that reinforcement learning policy and the cyclic training is really helping to reason on the temporal axis so that it can select the most important frames. Very interesting work.

**Weaknesses:**

The major weaknesses of this work are the following:

1 - It would be nice to see how the method affects other VLMs which are not flexible on the resolution/quality of input data.

2 - While the method performs really well in one of the most challenging problems of video understanding, it lacks comparison in more generic tasks like answer generation (for Q&A and captioning for example) to the the impact of this 'specialization' on other capabilities of the network.

3 - The ablation on cyclic training maybe needs a bit more deepening on the diminishing returns with additional cycles. The authors claim that it is due to the imperfect nature of MLLMs to provide correct answers for each visual scene and the limited info contained in 8 frames. I think these claims might need some more experimenting to see if the limit is on the method or on the MLLMs serving as supervisors.

**Questions:**

Check above.

---

> ### Author Response · Authors · 2025-11-21
>
> **W1**: It would be nice to see how the method affects other VLMs which are not flexible on the resolution/quality of input data ...
>
> **A**: We replace M1 with **InternVL3.5-4B**, which is not flexible with respect to input resolution, while keeping M2 as Qwen2.5-VL-7B. The setting is to select 8 frames from 64 frames, and we use an input resolution of 224×224. Note that InternVL3.5-4B’s original resolution is 448×448. Through experiments, we find that InternVL3.5-4B loses basic visual understanding capability when using the 112×112 resolution, whereas 448×448 or 128 frames causes out-of-memory errors. After completing 2 cycles, we evaluate on Video-MME, and the results are shown below.
>
> |     | short    |  medium   |  long   |   Avg  |
> |-----:|-----|-----|-----|-----|
> |  Qwen2.5-VL-7B   |  61.7   |  50.6   |  46.3   |  52.9   |
> |  InternVL3.5-4B + Qwen2.5-VL-7B   |66.9|55.9|50.0|57.6|
>
> InternVL3.5-4B also improves with our method, demonstrating its ability to generalize to models that are less flexible with respect to input resolution.
>
> **W2**: ... it lacks comparison in more generic tasks like answer generation (for Q&A and captioning for example) to the the impact of this 'specialization' on other capabilities of the network.
>
> **A**: We randomly select 8k open-ended QA samples--i.e., answer generation--from the LLaVA-Video-178K dataset and perform a single-step optimization of the selector(Cycle1-Stage1). The selector identifies 8 frames from the 128-frame input.
>
> We then evaluate the model using the Video-based Text Generation Performance Benchmark introduced in Video-ChatGPT [1], which has 1996 open-ended QA pairs.
> The results are shown below:
>
> | Model             | Correctness of Information | Detail Orientation | Contextual Understanding |
> | ----------------- | -------------------------- | ------------------ | ------------------------ |
> | Qwen2.5-VL        | 2.43                       | 2.58               | 3.06                     |
> | Qwen2.5-VL + Ours | 2.51                       | 2.62               | 3.10                     |
>
> We follow the evaluation setting of Video-ChatGPT, using GPT-3.5 to assess the response quality across several dimensions, such as Correctness of Information, Detail Orientation and Contextual Understanding.
> These results demonstrate that ViaRL continues to perform effectively on the open-ended QA task.
>
> **W3**: The ablation on cyclic training maybe needs a bit more deepening on the diminishing returns with additional cycles. The authors claim that it is due to the imperfect nature of MLLMs to provide correct answers for each visual scene and the limited info contained in 8 frames. I think these claims might need some more experimenting to see if the limit is on the method or on the MLLMs serving as supervisors.
>
> **A**: As shown in the second-to-last and fourth-from-last rows of Table 1, it is clear that for the answer model, providing more frames consistently leads to better performance.
> Therefore, when the answer model is provided with only 8 frames, the information available to it is naturally less than when more frames are supplied. Further it will affect the optimization of the selector.
> That's why we state in the paper that "the capabilities of using only 8 frames are inherently limited."
>
> Next, in ideal experimental setting, as long as the selected frames are appropriate, the answer model should theoretically achieve 100% accuracy.
> Nonetheless, currently there is no MLLM can provide such a guarantee, and even the strongest models cannot achieve such an ideal scenario.
> That's why we state in the paper that "there is currently no perfect multimodal large model capable of providing ideal answers based on the selected video frames all the time."
>
> Finally, we use the selector from Cycle2–Stage1 (3B model) and replace M2 with **Qwen2.5-VL-32B** as the answer model, which provides better answer reward. All other settings remain unchanged.
> We then conduct RL for the selector, and the results are shown below.
>
> |     | short    |  medium   |  long   |   Avg  |
> |-----|-----|-----|-----|-----|
> |  before RL  |63.2|55.1|51.9|56.7|
> |  after RL     |64.3|56.8|51.7|57.6|
>
> The results indicate that the selector remains improvable when paired with a stronger answer model.
>
> **References**:
>
> [1] Maaz, M., Rasheed, H., Khan, S., & Khan, F. Video-ChatGPT: Towards Detailed Video Understanding via Large Vision and Language Models. ACL 2024.

---

> > ### Comment · Reviewer_4Vp9 · 2025-11-21
> >
> > Thank you for the clarifications, I keep my score as it is.
> > Good luck!

---

> > > ### Author Response · Authors · 2025-11-24
> > > **Thanks for the recognition**
> > >
> > > Dear Reviewer 4Vp9, really appreciate your effort and recognition of our work. We are happy the rebuttal addressed all your questions.
> > >
> > > Best
> > > Authors

---

> ### Author Response · Authors · 2025-11-26
>
> Dear Reviewer 4Vp9, we have updated our response regarding the weaknesses 1&3 and provided additional results. We hope this addresses your concern.
>
> We would greatly appreciate your review of our rebuttal and any further feedback you may have.

---

### Official Review · Reviewer_KBUt · 2025-11-01

**Soundness:** 2
**Presentation:** 2
**Contribution:** 2
**Rating:** 4
**Confidence:** 5

**Summary:**

This paper introduces ViaRL (Visual Iterated Amplification Reinforcement Learning), a novel framework to address the challenge of efficient, query-driven frame selection in long videos. The authors argue that direct video reasoning in MLLMs is less effective than first mastering temporal grounding . The framework uses a cyclic, two-stage "Visual Iterated Amplification" training strategy: first, the Selector is improved via RL, and second, the Answer Model is instruction-tuned using the improved Selector's frame selections, creating a feedback loop where both models progressively enhance each other. Experiments show significant gains, especially a nearly 15% improvement on the Needle QA temporal grounding benchmark

**Strengths:**

- The paper is well-written and easy to follow.
- The training details are explained in detail. It improves the reproducibility of the paper.

**Weaknesses:**

- Unclear Inference Cost: The paper motivates its approach by citing the high cost of processing all frames. However, the proposed ViaRL framework requires two sequential MLLM forward passes at inference time
- From my perspective, the proposed paper lacks the technical novelty. Compared to existing GRPO-based works, the different part is to introduce frame selection before question answering. However, there have been multiple works that solve question-answering tasks with the frame selection.
- I'd like to see the contribution of the frame selection trained by reinforcement learning with verified rewards. It would be better if the paper included the performance comparison of the frame selection with other frame selection methods and the impact of them to the downstream tasks.

**Questions:**

- The current reward system is fundamentally tied to MCQ benchmarks . I wonder if the proposed ViaRL still works well on the open-ended generative tasks such as open-ended QA.

---

> ### Author Response · Authors · 2025-11-21
>
> **W2**: From my perspective, the proposed paper lacks the technical novelty. Compared to existing GRPO-based works, the different part is to introduce frame selection before question answering ...
>
> **A**: Actually, introducing frame selection before question answering is not the  distinctive aspect in our paper.
>
> As we all know, o3 provides a paradigm for reasoning within images, maintaining visual information throughout the process and focusing on relevant regions. Inspired by it, we explore how to identify the most relevant frames to a given query in a video sequence.
>
> While there are several frame-selection methods [1,2,3], all of them lack a clear training goal for assessing the quality of selected frames in the Video-CoT pipeline, primarily because the task is inherently subjective. Consequently, these methods can't be optimized through existing GRPO-based techniques.
>
> Therefore, we propose ViaRL which delegates the optimization of frame selection to a downstream MLLM. Besides, ViaRL uses the answer accuracy as a reward signal. This enables a trial-and-error learning process, eliminating the need for expensive frame selection annotations like pseudo labels or human-annotated labels.
>
> In fact, this method for optimizing selected frame quality is proposed for the first time.
>
> **W1**: Unclear Inference Cost: The paper motivates its approach by citing the high cost of processing all frames ...
>
> **A**: In the test of selecting 8 frames from 128 on Video-MME, we use the time relative to directly performing question answering on 128 frames as the metric. Our method takes 75% of the time required to process all frames, with the selector accounting for 57% and the answer model for 18%.
> It indicates that this method is more efficient than processing all frames.
>
> **W3**: ... It would be better if the paper included the performance comparison of the frame selection with other frame selection methods and the impact of them to the downstream tasks.
>
> **A**: Actually, we have a comparison in Fig 3 and discuss Sec 4.2, "Comparison of Frame Selection Models".
>
> **Q**: ... I wonder if the proposed ViaRL still works well on the open-ended generative tasks such as open-ended QA.
>
> **A**: We randomly select 8k open-ended QA samples from the LLaVA-Video-178K dataset and perform a single-step optimization of the selector(Cycle1-Stage1). The selector identifies 8 frames from the 128-frame input.
> We then evaluate the model using the Video-based Text Generation Performance Benchmark introduced in Video-ChatGPT [4], which has 1996 open-ended QA pairs.
> The results are shown below:
>
> | Model             | Correctness of Information | Detail Orientation | Contextual Understanding |
> | ----------------- | -------------------------- | ------------------ | ------------------------ |
> | Qwen2.5-VL        | 2.43                       | 2.58               | 3.06                     |
> | Qwen2.5-VL + Ours | 2.51                       | 2.62               | 3.10                     |
>
> We follow the evaluation setting of Video-ChatGPT, using GPT-3.5 to assess the response quality across several dimensions, such as Correctness of Information, Detail Orientation and Contextual Understanding.
> These results demonstrate that ViaRL continues to perform effectively on the open-ended QA task.
>
>
>
> **References**:
>
> [1] Tang, X., Qiu, J., Xie, L., et al. Adaptive keyframe sampling for long video understanding. CVPR 2025.
>
> [2] Hu, J., Cheng, Z., Si, C., et al. Cos: Chain-of-shot prompting for long video understanding. arXiv:2502.06428.
>
> [3] Hu, K., Gao, F., Nie, X., et al. M-LLM based video frame selection for efficient video understanding. CVPR 2025.
>
> [4] Maaz, M., Rasheed, H., Khan, S., & Khan, F. Video-ChatGPT: Towards Detailed Video Understanding via Large Vision and Language Models. ACL 2024.

---

> > ### Author Response · Authors · 2025-11-24
> > **Help read the rebuttal and provide feedback**
> >
> > Dear reviewer KBUt, we would greatly appreciate it if you could review our rebuttal and share your feedback. Please let us know if further clarification or additional responses are needed.

---

### Official Review · Reviewer_8prC · 2025-11-01

**Soundness:** 3
**Presentation:** 2
**Contribution:** 3
**Rating:** 4
**Confidence:** 4

**Summary:**

This paper introduced ViaRL, a framework to use rule-based RL to optimize the frame selection process in video understanding tasks. It is an iteration process is used in the CoT process and ViaRL use the accuracy of downstraming model as the reward signal. In more detail, they first processed a video understanding dataset by using CLIP to sample frames based on visual-textual similarity and then filter out less informative tasks. They using reinforce++ to finetune Qwen model to select frames and using another answer model to provide training signal. Each model is tuned in turn to help each other.
They demonstrated good performance on popular video understanding benchmark including VideoMME, LVBench, and MLVU.

**Strengths:**

1. This paper provided a RL framework for temporal grounding without human annotation. By iteratively refine the selector and the answer model on the training set, the performance of Qwen is improved.
2. The paper also provided useful tricks. For example, using idea from existing work to mark the frame index in the frame corner.
3. The author achieved great performance on Qwen model.

**Weaknesses:**

1. The RL is only tested on Qwen-2.5-VL, therefore it is hard to know if the method could generalize to other models.
2. In the data preparation process, CLIP is used to sample relevant frame to the question, which could be inaccurate. CLIP is measuring the semantic similarity between the frame and the answer, while a frame could be barely connected with the question when it is alone but important when in frames context. I also do not see any experiments supporting this sampling process.
3. Table 2 is never discussed in the paper. For example, why using all components yield the worse performance among the table?
4. The author mentioned several RL rewards but only length reward is analyzed. Then it is not clear why the author use those rewards. Although they are proven useful in other tasks in other paper, it is not clear whether they help in this case.

**Questions:**

1. In table 1, the bold number only represents higher number between Qwen and Qwen+ViaRL, but the authors did not mention this. The bold results are worse than many open-source MLLMs and Proprietary Models. This is a minor issue so I put it in questions section. Hope the authors could clarify this in the paper.

2. Why choosing Qwen-7B as the answer model? Here I have several questions. 1. After improving, will the answer model achieve sota performance? If not, why not using existing sota model (open sourced or not) to provide the reward?

---

> ### Author Response · Authors · 2025-11-21
>
> **Q1**: In table 1, the bold number only represents higher number between Qwen and Qwen+ViaRL, but the authors did not mention this...
>
> **A**: Thanks for the reminder. We provide a method that helps existing models further explore their capabilities and improve their performance like a plug-and-play manner, rather than training a new SOTA model.
> Therefore, it's more important to compare the method (Qwen+ViaRL) with the basic model (Qwen) to demonstrate the effectiveness. This comparison strategy has also been adopted in several works, like AKS[1] and CoS[2].
>
> **Q2**: Why choosing Qwen-7B as the answer model? ...
>
> **A**: As we clarified in Q1, the answer model will achieve a better performance after improving, and the effectiveness of our method has been demonstrated.
> As for choosing Qwen-7B, it offers good performance while its 7B size allows for fine-tuning locally. Some SOTA models are either closed-source and cannot be fine-tuned, or open-source but too large to be feasible with available computational resources.
>
> **W1**: ... if the method could generalize to other models.
>
> **A**:
> We replace M1 with **InternVL3.5-4B**, while M2 remains Qwen2.5-VL-7B. The setting is to select 8 frames from 64 frames, and we use an input resolution of 224×224. Note that InternVL3.5-4B’s original resolution is 448×448. Through experiments, we find that InternVL3.5-4B loses basic visual understanding capability when using the 112×112 resolution, whereas 448×448 or 128 frames causes out-of-memory errors. After completing 2 cycles, we evaluate on Video-MME, and the results are shown below.
>
> |     | short    |  medium   |  long   |   Avg  |
> |-----:|-----|-----|-----|-----|
> |  Qwen2.5-VL-7B   |  61.7   |  50.6   |  46.3   |  52.9   |
> |  InternVL3.5-4B + Qwen2.5-VL-7B   |66.9|55.9|50.0|57.6|
>
> InternVL3.5-4B also improves with our method. Besides, as shown in Table 3, we have also tested the MiMo-VL-7B, which acts as a selector, and obtained a similar improvement.
>
> Both MiMo-VL-7B and InternVL3.5-4B show improvements with our method, demonstrating its ability to generalize to other models.
>
> **W2**: In the data preparation process, CLIP is used to sample relevant frame to the question, which could be inaccurate ...
>
> **A**: Actually, our view is consistent with yours. For the selection of RL data, we filtered out instances where the question could be answered using the relevant frames selected by CLIP, in order to ensure that the data is more challenging. This is described in detail in Appendix B.
>
> **W3**: Table 2 (Table 3 in the new version) is never discussed in the paper ...
>
> **A**: It is discussed in Section 4.3, “Different Training Recipes”. This is likely due to the article layout, and we will adjust the placement of table and text in the final version.
>
> **W4**: The author mentioned several RL rewards but only length reward is analyzed ...
>
> **A**: Rewards such as answer, format and frame index are essential for training and training can't proceed without them. Answer is the most crucial reward for optimizing the selector, and training is impossible without it. Without the format or frame index reward, it is difficult to obtain $N$ distinct frame indices in the correct format from the selector’s output throughout training, and as a result, the answer model cannot receive valid video frame inputs. This would make the training ineffective.
> Besides, we have a discussion about accuracy reward in Sec 4.3, "Sensitivity Analysis on the Answer Score", and Fig 4.
>
>
> **References**:
>
> [1] Tang, X., Qiu, J., Xie, L., et al. Adaptive keyframe sampling for long video understanding. CVPR 2025.
>
> [2] Hu, J., Cheng, Z., Si, C., et al. Cos: Chain-of-shot prompting for long video understanding. arXiv:2502.06428.

---

> > ### Author Response · Authors · 2025-11-24
> > **Help read the rebuttal and provide feedback**
> >
> > Dear Reviewer 8prC, we would greatly appreciate it if you could review our rebuttal and share your feedback. Please let us know if further clarification or additional responses are needed.

---

> ### Author Response · Authors · 2025-11-26
>
> Dear Reviewer 8prC, we have updated our response regarding the “Weaknesses: ... if the method could generalize to other models” and provided additional results demonstrating the effectiveness of our approach. We hope this addresses your concern.
>
> We would greatly appreciate your review of our rebuttal and any feedback you may have.

---

### Meta-Review · Area_Chair_FzFe · 2026-01-10

**Summary:**

Reviewers are divided on this submission. **Reviewer 8prC** and **Reviewer KBUt** recommend rejection, while **Reviewer 4Vp9** argues for acceptance.

**Reviewer 8prC** identifies several weaknesses in the evaluation. The original submission relied on a single backbone and used CLIP-based pre-filtering. This reviewer also points out a lack of clear reward analysis and notes that Qwen+ViaRL still lags behind existing MLLMs. They question if Qwen-7B can ever reach SOTA performance levels.

**Reviewer KBUt** questions the technical novelty. They see the work as too similar to GRPO and existing frame-selection methods. They also highlight the high inference cost of a two-stage pipeline. The paper lacks comparisons to other frame selection baselines and relies on rewards tied to multiple-choice QA. This makes it unclear if the method works for open-ended tasks.

**Reviewer 4Vp9** is the most positive. They find the visual iterated amplification idea sound and appreciate the performance gains. However, they also requested more validation on different VLMs and tasks beyond QA.

The authors added experiments on InternVL3.5-4B and MiMo-VL-7B during the rebuttal. They explained that CLIP filtering removes trivial cases and argued that all reward components are necessary. They see ViaRL as a plug-and-play enhancement rather than a new SOTA system. These responses clarify some points but do not fix the core issues. The technical novelty remains limited and the system still trails stronger baselines. I do not believe the paper meets the bar for acceptance.

**Reviewer Concerns:**

**Reviewer 8prC** focuses on generalization and the role of the reward design.

The original submission focused on Qwen-2.5-VL and gave little evidence that the method works on other backbones. This reviewer also worries that CLIP-based sampling might miss task-specific details. The gains in Table 2 mostly compare Qwen against Qwen+ViaRL, and the final system remains weaker than many MLLMs.

The authors addressed some of these points in the rebuttal. They added InternVL3.5-4B and MiMo-VL-7B to show generalization. They clarified that CLIP is not used to define frame importance. However, the rebuttal does not explain why the full reward set sometimes underperforms its subsets in ablations. It also fails to show that ViaRL closes the gap to stronger models. The concerns about empirical impact remain.

**Reviewer KBUt** focuses on novelty, cost, and missing baselines.

They see limited technical novelty compared to GRPO-based methods. The two-pass pipeline creates high inference costs, and the paper omits direct comparisons to existing frame selection work. This reviewer also questions the narrow focus on multiple-choice QA.

In the rebuttal, the authors framed ViaRL as a practical training framework. They justified the Qwen-7B choice through training constraints. This explains the setup but does not fix the two-pass cost structure. The authors did not add new frame-selection baselines or open-ended generation results. Because of this, the doubts about novelty and applicability remain.

**Reviewer 4Vp9** supports the work and finds the core idea interesting.

They want to see how the method performs on VLMs with lower resolution or visual quality. They also asked for tests on open-ended QA and captioning. A more detailed study of training cycles was also requested to check for diminishing returns.

The authors provided results on InternVL3.5-4B and MiMo-VL-7B during the rebuttal. This helps show the framework is flexible. But they did not include results for general generation tasks. They also did not offer a deep analysis of cycle counts. While this reviewer remains positive, several of their requests for broader evidence were not fully met.

**Reviewer Scores:**

- **Reviewer 8prC (Original: 4 → Predicted: 4)**
  The rebuttal adds some models but the empirical results are still limited. Concerns about the performance gap and the reward design are not fully resolved. The score will likely stay at 4.

- **Reviewer KBUt (Original: 4 → Predicted: 4)**
  This reviewer highlighted technical novelty and missing baselines. The rebuttal mostly explained the current setup rather than adding new comparisons. Their core concerns remain, so the score will likely stay at 4.

- **Reviewer 4Vp9 (Original: 8 → Predicted: 8)**
  This reviewer likes the concept. The authors partially addressed their requests for more backbones. Since the initial impression was strong, the score will likely stay at 8.

---

### Decision · Program_Chairs · 2026-01-26

Reject